# A prospective, phase II, single-centre, cross-sectional, randomised study investigating Dehydroepiandrosterone supplementation and its Profile in Trauma: ADaPT

Conor Bentley [1,2] Claire Potter,[1,3] Kamal Makram Yakoub [1] Kristian Brock,[3] Victoria Homer,[3] Emma Toman [1] Angela E Taylor,[4] Fozia Shaheen,[4] Lorna C Gilligan,[4] Amrita Athwal,[3] Darren Barton,[3] Ronald Carrera,[1] Katie Young,[1] Amisha Desai,[1] Kirsty McGee,[5] Christos Ermogenous,[1] Gurneet Sur,[3,6] Carolyn A Greig,[2,7] Jon Hazeldine [1,5] Wiebke Arlt,[4,8] Janet M Lord [1,5,8] Mark A Foster[1,9]

**Correspondence to**
Conor Bentley;
conor.bentley@nhs.net

## ABSTRACT

**Introduction** The improvements in short-term outcome after severe trauma achieved through early resuscitation and acute care can be offset over the following weeks by an acute systemic inflammatory response with immuneparesis leading to infection, multiorgan dysfunction/multiorgan failure (MOF) and death. Serum levels of the androgen precursor dehydroepiandrosterone (DHEA) and its sulfate ester DHEAS, steroids with immune-enhancing activity, are low after traumatic injury at a time when patients are catabolic and immunosuppressed. Addressing this deficit and restoring the DHEA(S) ratio to cortisol may provide a range of physiological benefits, including immune modulatory effects.

**Objective** Our primary objective is to establish a dose suitable for DHEA supplementation in patients after acute trauma to raise circulating DHEA levels to at least 15 nmol/L. Secondary objectives are to assess if DHEA supplementation has any effect on neutrophil function, metabolic and cytokine profiles and which route of administration (oral vs sublingual) is more effective in restoring circulating levels of DHEA, DHEAS and downstream androgens.

**Methods and analysis** A prospective, phase II, single-centre, cross-sectional, randomised study investigating Dehydroepiandrosterone supplementation and its profile in trauma, with a planned recruitment between April 2019 and July 2021, that will investigate DHEA supplementation and its effect on serum DHEA, DHEAS and downstream androgens in trauma. A maximum of 270 patients will receive sublingual or oral DHEA at 50, 100 or 200 mg daily over 3 days. Females aged ≥50 years with neck of femur fracture and male and female major trauma patients, aged 16–50 years with an injury severity score ≥16, will be recruited.

**Ethics and dissemination** This protocol was approved by the West Midlands – Coventry and Warwickshire Research Ethics Committee (Reference 18/WM/0102) on 8 June 2018. Results will be disseminated via peer-reviewed

## Strengths and limitations of this study

► Identify the dose and route of administration needed to normalise dehydroepiandrosterone levels in a cohort that are known to have low levels, both habitually and post-traumatic injury.

► The rapid turnaround time from bedside to bench and bench to interim analysis will minimise inappropriate dosing and public money expenditure on ineffective doses.

► This swift analysis will limit patient exposure to insufficient dosing and unnecessary specimen collection.

► A study limitation could be its single-site design; however, this will facilitate collecting sensitive immunological samples.

publications and presented at national and international conferences.

**Trial registration** This trial is registered with the European Medicines Agency (EudraCT: 2016-004250-15) and ISRCTN (12961998). It has also been adopted on the National Institute of Health Research portfolio (CPMS ID:38158).

**Trial progression** The study recruited its first patient on 2 April 2019 and held its first data monitoring committee on 8 November 2019. DHEA dosing has increased to 100 mg in both male cohorts and remains on 50 mg in across all female groups.

## INTRODUCTION

The improvements in short-term outcome after severe trauma achieved through early resuscitation and acute care are offset over the following weeks by an acute systemic inflammatory response with immuneparesis leading

to infection, multiorgan dysfunction/multiorgan failure (MOF) and death.[1 2] The combination of excessive pro-inflammatory and anti-inflammatory responses impairs the rehabilitation phase of trauma, including wound healing, physical and emotional recovery.[3] Upregulation of glucocorticoid biosynthesis promotes a catabolic state, lasting several weeks and is associated with a significant reduction in muscle mass.[4]

Analysis of gender differences in trauma has shown that women, particularly those under 30 years of age, have fewer infections and a better outcome from sepsis than men.[5 6] The protective effects of oestrogens on immune cells and organ function highlight the potential role of sex hormones in modulating trauma-related immune dysfunction.[7] Cellular immunity is also influenced by hormone production, and members of our group have shown that the adrenal response to stress, specifically the ratio of cortisol to dehydroepiandrosterone sulfate (DHEAS), is linked to neutrophil bactericidal function, specifically superoxide production.[8 9]

Dehydroepiandrosterone (DHEA) and its sulfate ester DHEAS are the most abundant steroids in the human circulation; DHEA is a precursor of sex hormones modulating several physiologic processes including metabolism, muscle protein synthesis and cardiovascular function.[10 11] DHEA is converted to active androgens in peripheral target cells including immune cells[12] and is also converted to DHEAS by the action of DHEA sulfotransferase (SULT2A1).[13] We have shown that DHEA sulfation is downregulated in acute systemic inflammatory response syndrome (SIRS) and sepsis.[14] Hazeldine et al[15] reviewed the numerous immune effects of DHEA highlighting how DHEAS, but not DHEA, enhances neutrophil superoxide generation via a protein kinase C mediated pathway, a vital immune response in fighting bacterial infections.[8]

The roles of DHEA and DHEAS in severe injury are relatively unexplored. The majority of studies focus on cortisol responses,[16] whereas our data suggest that it is the cortisol:DHEAS ratio post-trauma that has a superior prognostic ability.[17 18] Although critical for survival, prolonged hypercortisolaemia is known to be detrimental in part due to its immunosuppressive effects.[19] This intra-adrenal shift causes decreased levels of circulating testosterone and oestrogen, resulting in rapid lean body mass loss,[20] in addition to increased susceptibility to infection and sepsis.[21] Correcting the cortisol:DHEA or cortisol:DHEAS ratio via the administration of DHEA has yet to be undertaken in traumatically injured patients despite DHEA and DHEAS being below the reference ranges for 6 weeks and 6 months postinjury, respectively.[4]

DHEA is subject to first-pass metabolism in healthy individuals, which in turn may lead to a non-physiological metabolism after an oral dose.[22] Bypassing first-pass metabolism using a sublingual or buccal preparation[23] should improve the bioavailability of DHEA.[24 25] Previous studies in healthy older people have shown that supplementation with 50 mg DHEA orally once daily can restore both serum DHEA and DHEAS levels to that of men and women in the third decade of life.[22 26] Furthermore, the current literature suggests that DHEA supplementation may be beneficial for immune function and could extend to bone remodelling, muscle composition, psychological and neurological improvements.[27]

This study will seek to estimate the optimal dose and route of administration of DHEA in trauma patients and increase serum levels of DHEA and DHEAS to those seen in healthy adults. We also aim to find the optimal dose to enhance immune function, as demonstrated through changes in neutrophil phagocytosis and reactive oxygen species (ROS) production. The pilot data generated from the study is necessary to develop the protocol for a randomised controlled trial that will determine whether DHEA supplementation may improve outcomes in injured patients.

## RATIONALE
### Justification of the patient population
Despite improvements in short-term outcomes from traumatic injury via aggressive early resuscitation and acute care, over 5 million people worldwide die each year from serious injury.[28] With improved prehospital medicine mitigating the immediate threats to life it is the following weeks after traumatic injury that has seen the dysregulated SIRS in susceptible patients. The SIRS response is accompanied by a compensatory anti-inflammatory response (CARS)[29] to restore homeostasis. SIRS and CARS may progress to the persistent inflammation, immunosuppression and catabolism syndrome (PICS).[30 31] PICS further compounds the insult from the initial injury and results in an increased risk of infection, MOF and late deaths.[1 2] Therefore, strategies to mitigate these detrimental outcomes for patients are needed in the short-term, medium-term and long-term care of trauma patients.

The young trauma cohort will be recruited alongside a cohort of older (≥50 years old) female patients who have sustained a hip fracture at the neck of femur (NOF). According to the National Hip Fracture Database (2018), the NOF burden on the UK economy is estimated to be around £1 billion per annum.[32] This considerable cost is set to rise as the population ages. By the age of 40 years, decreasing serum levels of DHEA and DHEAS are observed in both sexes,[33 34] a phenomenon sometimes referred to as 'adrenopause'. Circulating levels of DHEA and DHEAS are lower in women than in men; however, in postmenopausal women, adrenal androgens are a source of almost all active oestrogens.[35] To the best of our knowledge, there has been no traumatic injury or NOF interventional studies supplementing DHEA or DHEAS, despite its therapeutic potential in the immediate to longer term care of the young and aged trauma patient. In this pilot study, we will recruit a female NOF cohort as well as a young adult trauma cohort with both male and female patients presenting at the major trauma

centre (MTC) and intensive care unit (ICU) at Queen Elizabeth Hospital, Birmingham (QEHB), UK.

## Choice of treatment

The DHEA doses chosen in this study (50 mg, 100 mg and 200 mg) were selected based on in vivo studies that demonstrated these doses to be both safe and effective at raising the levels of DHEA to that of healthy young adult levels.[36] DHEA is at its highest in the third decade of life. After which there is a steady decrease over the life course in both males and females. These doses have also previously been used in adrenal insufficiency,[37] older people[38] and young males[39] and females,[23] causing transient hyperandrogenism with acne being the most frequently reported side effect.[40 41] DHEA is activated to downstream androgens by stepwise conversion catalysed by several steroidogenic enzymes. However, we do not know whether the expression and activity of these enzymes are affected by major trauma and inflammation. There is evidence that inflammation and trauma affect DHEA sulfation[4 14] and may shift the conversion of DHEA to a higher rate of androgen activation.

Trauma and inflammation can impact adversely on gut absorption[42] and hepatic first-pass metabolism.[43] Therefore, we have chosen to administer DHEA as a sublingual preparation. Sublingual administration is commonly used by nursing staff in the ICU context, especially in those patients who have contraindications to oral administration or by patients wishing to self-administer.[44]

## OBJECTIVES

### Primary objective

The primary objective of this study is to establish the daily dose of DHEA that restores serum DHEA levels to at least 15 nmol/L, that is, the mid-healthy adult reference range, in trauma and hip fracture patients after 3 days of supplementation.

### Secondary objective

Secondary objectives include observing the effect of single and multiple DHEA doses on circulating DHEA, DHEAS and downstream androgens. Additionally, we will investigate whether the route of DHEA administration (oral vs sublingual) modulates the profile of circulating DHEA. This will be determined by assessing the efficacy of each route via statistical modelling. The immune response to the DHEA supplementation will be assessed as a marker of potential clinical relevance.

## METHODS

### Trial design

A prospective, phase II, single-centre, cross-sectional, randomised study investigating Dehydroepiandrosterone supplementation and its Profile in Trauma (ADaPT) trial conducted in male and female adult trauma patients and older females who have suffered neck of the femur fracture. This is a single-site study with patients recruited from QEHB. Three doses of DHEA will be investigated in this trial: 50, 100 and 200 mg, each administered once daily for 3 days via either oral or sublingual tablets. The trial has an adaptive design in order to answer both the primary and secondary objectives, with regular interim analysis to minimise the investigation of inactive doses. The trial consists of two components. The first component of the trial is to determine the dose of DHEA needed to sufficiently raise serum DHEA levels to at least 15 nmol/L after 3 days of DHEA administration. Based on previous work, 15 nmol/L has been selected as this is the lower acceptable level of DHEA observed in healthy young adults. Our recent analysis of the steroid response to trauma has shown that DHEA levels were very low and often undetectable for several weeks after trauma.[4] The second component of the trial is to investigate if DHEA will enhance neutrophil function. The study was approved by The Medicines and Healthcare products Regulatory Agency (MHRA) for the use of DHEA as an investigational medicinal product (IMP). Ethical approval was obtained from the West Midlands Research Ethics Committee (Reference 18/WM/0102). The trial will be conducted in accordance with the Declaration of Helsinki (World Medical Association 2008). Figure 1 summarises the patient pathway of the trial. The protocol (V.5.0, 28 June 2019) has been prepared in accordance with the Standard Protocol Items: Recommendations for Interventional Trials (SPIRIT) guidelines.[45]

### Patient selection

A maximum of 270 patients will be enrolled across three patient groups (young male trauma, young female trauma and female hip fracture). These trauma patient groups have been selected due to the immuneparesis effects caused by the acute systemic inflammatory response that follows severe trauma. The hip fracture group was selected as they have low serum DHEA and DHEAS due to adrenopause, and there are several papers showing an association between HPA axis activity measures and outcomes in these patients.[9 18 46 47] Patients admitted to QEHB will all be prescreened and assessed for eligibility. Patients will be screened between 07:00 and 20:00, 7 days a week. Potential participants will have an assessment of their medical history and current trauma injury, and the eligible patients will be recruited. The eligibility criteria have been split into trauma patients and hip fracture patients (box 1). The study will not exclude NOF patients with dementia where supplementation is currently being tested in the prevention and treatment of age-related cognitive impairment without deleterious effects.[48]

### Randomisation

Patients who meet the eligibility criteria and provide consent are randomised to receive DHEA via either oral tablets or sublingual tablets once daily for 3 days using a 1:1 randomisation ratio. With 3 populations of patients (male trauma, female trauma and female hip fracture)

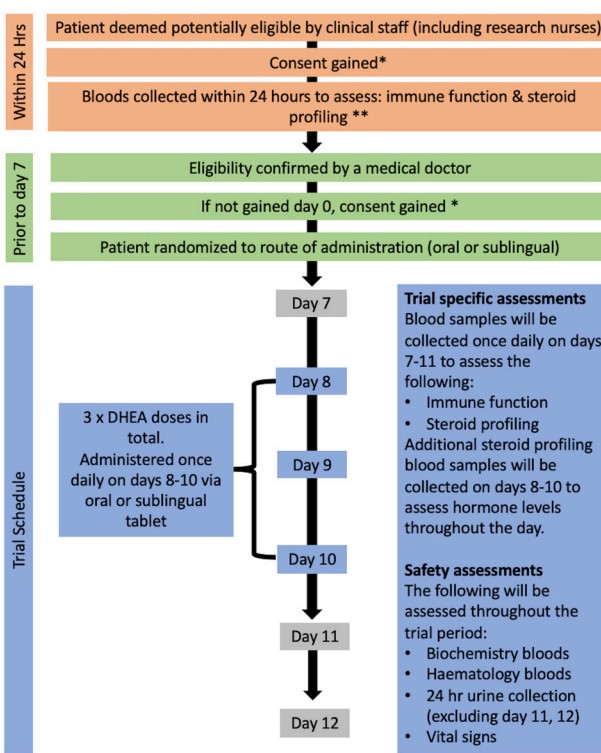

**Figure 1** A trial flow chart describing the patient journey in through the ADaPT study. * Due to the nature of the injury, informed consent can be sought from a professional legal representative or personal legal representative if the patient does not have capacity. Consent from the patient will be obtained at the earliest opportunity by research team members. **Twenty-four-hour blood and consent will only be collected within 24 hours of injury. The omission of this sample collection does not render a patient unevaluable. ADaPT, A prospective, phase II, single-centre, cross-sectional, randomised study investigating dehydroepiandrosterone supplementation and its Profile in Trauma; DHEA, dehydroepiandrosterone.

and two routes of administration, there will be six groups in total. Randomisation will take place using a secure web-based tool. Nursing and medical staff will use the Clinical RESearch Tool (CREST), developed at University Hospitals Birmingham Foundation Trust (UHB), to randomly assign patients and provide an anonymised electronic case report form (CRF), for trial management, data collection and adverse event reporting. The prevailing dose of DHEA (50, 100 or 200 mg) for administration in a group will be adapted in response to sequential analysis of interim outcomes. Each group will have its own dose. Blinding is not possible as the difference in DHEA delivery method is evident to both the clinician and the participant. If a contraindication to oral or the sublingual route present prior to commencing the study intervention, forced randomisation will occur.

### Study intervention
The supplementation of DHEA will commence in the second week after injury, which has previously been

---

**Box 1  Patient inclusion and exclusion criteria**

**Trauma patients inclusion criteria:**
► Aged 16–50 years of age.
► Severely injured trauma patient (injury severity score (ISS) ≥16 and ≤50).
► Admitted to University Hospital Birmingham within 6 days of trauma.
► Anticipated to be an inpatient for the 12-day trial period.

**Trauma patients exclusion criteria:**
► Ages <16 or >51.
► ISS <16 or >50.
► Isolated brain injury.
► Unlikely to survive the study period.
► Known hormone sensitive malignancy.
► Known prostatic hypertrophy (M).
► Female patients taking hormone replacement therapy (HRT) medication or oral contraceptives.
► Intake of any drugs that interfere with adrenal function in the last 3 months
► Pre-existing liver impairment or chronic liver failure.
► Previous or current malignancy or invasive cancer diagnosed within the past 3 years except for adequately treated basal cell and squamous cell carcinoma of the skin and in situ carcinoma of the uterine cervix.
► Pregnant and/or breastfeeding females (women of childbearing potential to complete serum pregnancy test).
► Known hypersensitivity to the active substance or excipient.
► Known thromboembolic events in the last 12 months and any predisposition to thrombosis,e.g, Factor V Leiden.

**Hip fracture patients inclusion criteria**
► Aged 50 years and older.
► Female.
► Neck of femur fracture.
► Admitted to University Hospital Birmingham within 6 days of fracture.
► Anticipated to be an inpatient for the 12-day trial period.

**Hip fracture patients exclusion criteria**
► <50 years old.
► Unlikely to survive the study period.
► Previous or known hormone sensitive malignancy.
► Intake of any drugs that interfere with adrenal function in the last 3 months.
► Pre-existing liver impairment or chronic liver failure.
► Previous or current malignancy or invasive cancer diagnosed within the past 3 years except for adequately treated basal cell and squamous cell carcinoma of the skin and in situ carcinoma of the uterine cervix.
► Pregnant and/or breastfeeding (women of childbearing potential to complete serum pregnancy test).
► Known hypersensitivity to the active substance or excipient.
► Females on HRT medication.
► Known thromboembolic events in the last 12 months and any predisposition to thrombosis e.g. Factor V Leiden.

shown to be a time when trauma patients become maximally unwell as a result of sepsis and MOF.[49] Additionally, this time point has been selected to optimise patient recruitment from both the NOF cohort and trauma patients (median stay 18 days and 13 days, respectively) both DHEA and DHEAS levels post-injury.[50] Recruiting

---

an in-hospital cohort provides an opportunity to monitor patients and assess the impact that this class C controlled drug has on the serial steroid profile and immune function during a period of vulnerability over 3 days of administration.

## Participant timeline

The trial intervention will occur over 3 days, during the first 12 days while patients are at QEHB. Three doses of DHEA will be investigated in all patients, and dosing will occur on days 8, 9 and 10 only. All cohorts will begin the study on 50 mg, and the dose administered to the next eligible patient to be included within a cohort will be escalated when interim analysis determines if the primary objective has been reached. At no point will any patient escalate once they have been allocated a dose of DHEA.

## Dose escalation

Dose escalation for the DHEA restoration part of the trial is dependent on the serum DHEA levels. A dose that restores serum DHEA to ≥15 nmol/L (referred to as 'normalise DHEA') is sought in at least 13 out of 15 patients or approximately 85% of patients. The decision to escalate dose in a cohort will be driven in the main by the outcomes of the patients in that cohort and escalation will occur independently for each cohort. However, valuable additional information will be garnered from the outcomes of patients in other cohorts. Flexible information sharing across related groups will be achieved using hierarchical regression models. The requirement for 13 out of 15 patients to normalise in the final effective dose is based on what was judged to be clinically important and thus would warrant investigation of the chosen dose of DHEA in a confirmatory phase III trial to ascertain superiority of outcomes compared with standard of care.

To minimise the investigation of inactive doses, we propose an analysis of each dose level in each cohort after 3 to 5 evaluable patients have been assessed. It will be taken as evidence that the normalisation rate is too low and that the dose should be increased if: any patient fails to normalise (where n=3) or more than one patient fails to normalise (where n=4 or 5). This rule-based analysis will provide an early opportunity to escalate in search of a more promising dose when a probabilistic analysis will likely be underinformed due to the small sample size. If the requisite number normalise, recruitment will continue in that cohort at that dose-level up to a maximum of 15.

Once an effective dose has been established to normalise DHEA levels within a group and at least 15 patients have completed it, the DHEA will be escalated to the next dose to satisfy the second component of the trial and to determine whether further increases in DHEA supplementation will enhance immune function. The immune response component will focus on neutrophil phagocytosis and ROS production that will involve fewer research samples. Fifteen patients (within a cohort) will complete the immune response part of the trial at each

subsequent dose escalation. If 50 mg is established to be sufficient to normalise DHEA, the dose will be escalated twice to investigate whether 100 mg or 200 mg is optimal for increasing the immune response. Both 100 mg and 200 mg have safely been used in previous studies,[51 52] but this has not been addressed in the context of trauma. Each group will be escalated independently of each other (figure 2).

## Patient and public involvement (PPI)

Before the beginning of the study, patients and lay members of the '1000 Elders' group at the University of Birmingham (UoB) were invited to group PPI sessions held by the Surgical Reconstruction and Microbiology Reconstruction Centre (SRMRC) at QEHB. During these interactive group sessions, discussions were held to showcase the work that was planned to be undertaken to address previously highlighted problems in their recovery from traumatic injury. Members of the group informally reviewed, developed and fed-back on the study design and patient paperwork; contemporaneous records were generated. On entry and active participation with the ADaPT study, patients were asked if they would like to become members of the PPI group and assist in the ongoing evaluation and future dissemination of the project. The participants will be asked to participate in a grant application should the results of this study warrant a more extensive phase III, multicentre trial.

## PRIMARY AND SECONDARY OUTCOMES

The primary outcome for the study is serum DHEA after 3 days of DHEA supplementation. Previous research into DHEA levels and DHEA supplementation use DHEAS as the primary endpoint for determining whether the supplementation has been effective at raising levels. However, DHEAS levels do not act as a proxy marker for DHEA levels in the trauma population.[50] Using liquid chromatography-tandem mass spectrometry (LC-MS/MS), we have shown that DHEA and DHEAS both behave differently after trauma injury.[53] Animal models of trauma have demonstrated improvements in hyperglycaemia,[54] decreased mortality after trauma-induced haemorrhage,[55] neurogenesis[56] and wound reperfusion.[57] Human studies including a recent meta-analysis suggested that DHEA supplementation may be beneficial in increasing bone mineral density[58] in women, increasing muscle strength,[59] improving mood in those with moderate depression[60] and adrenal insufficiency.[37]

These potential restorative immunological, physiological and psychological benefits seen in animals and human studies can only be investigated once the appropriate dose of DHEA to restore normal levels and the most suitable administration route has been identified. We know that supplementation of DHEA in healthy subjects via oral administration will result in significant first-pass metabolism and, thus, more extensive conversion of DHEA to DHEAS than is observed via, for example, transdermal

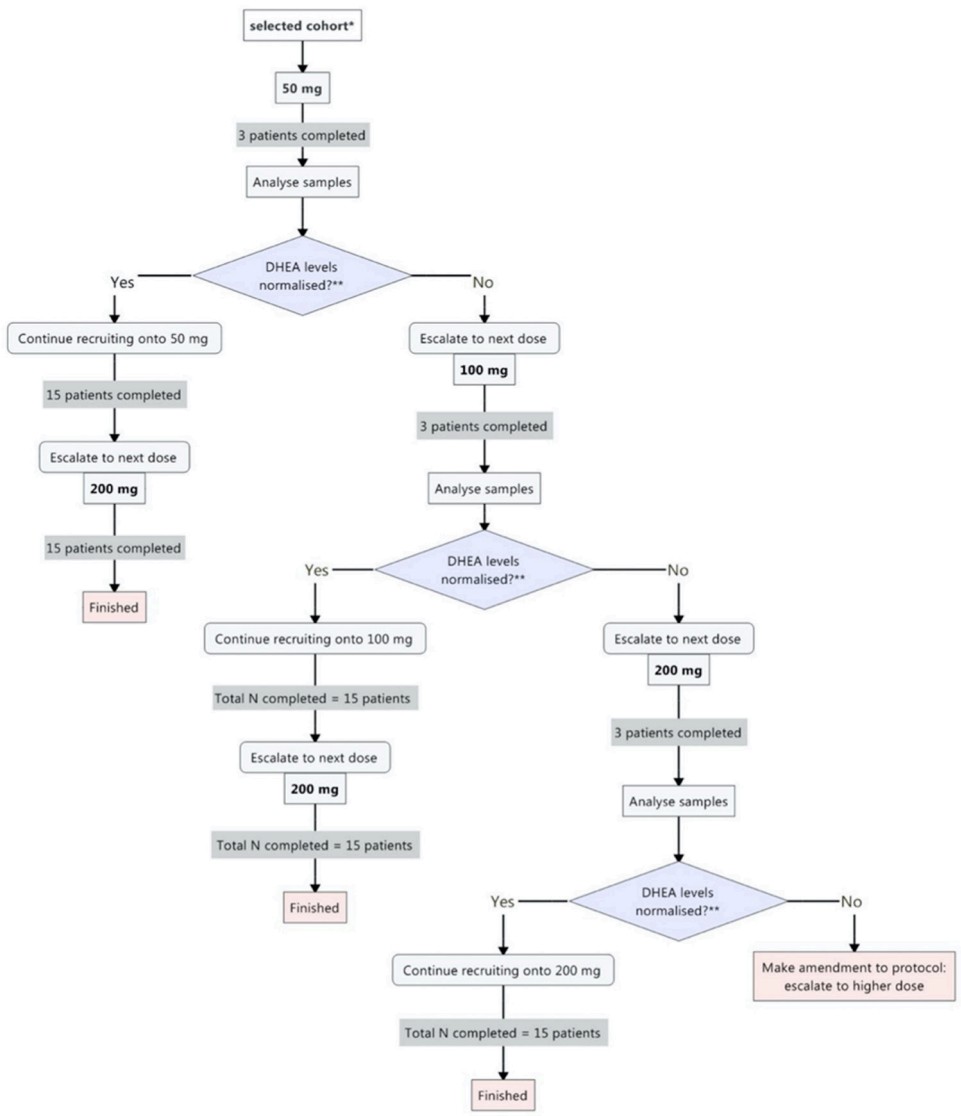

**Figure 2** Indicative flow chart to explain dose escalation design of the ADaPT trial. *Cohorts: oral male trauma, sublingual male trauma, oral female trauma, sublingual female trauma, oral female hip fracture and sublingual female hip fracture. ADaPT, .A prospective, phase II, single-centre, cross-sectional, randomised study investigating dehydroepiandrosterone supplementation and its Profile in Trauma; DHEA, dehydroepiandrosterone.

administration,[61] which is why we plan to compare oral versus sublingual DHEA administration.

One potential instantaneous benefit to trauma patients, which may be observed systemically after a dose of DHEA, is a positive effect on the bactericidal function of neutrophils[8] by enhancing ROS generation via activation of nicotinamide adenine dinucleotide phosphate (NADPH) oxidase.[17] Neutrophil function will therefore be investigated as a secondary outcome, with limited expectations on the results of the pilot data, given that DHEA will only be administered for 3 days.

### Steroid analysis

DHEA and downstream androgens will be quantified using a validated LC-MS/MS multisteroid profiling method carried out on a Waters Xevo-XS, with acquity uPLC, using a water/methanol (0.1% formic acid) gradient system and a HSS T3, 1.8 µm, 1.2×50 mm column.

Steroids are extracted via liquid–liquid extraction using tert-butyl methyl ether (MTBE), following the addition of an internal standard and protein precipitation using acetonitrile. The MTBE layer containing steroids was transferred and evaporated under nitrogen then reconstituted in 125 µL of 50/50 methanol/water before analysis. Steroids will be quantified against a calibration series ranging from 0.05 to 250 ng/mL.[62–66]

DHEAS was measured separately. Twenty microliters of serum was spiked with internal standard, followed by 100 µL of acetonitrile and 20 µL of $ZnSO_4$. The samples were then centrifuged, and 100 µL of the solution was transferred to a new vial, dried and reconstituted in 200 µL of methanol/water prior to LC-MS/MS analysis as described by Chadwick et al.[67] DHEAS will be quantified against a calibration series ranging from 250 to 8000 ng/mL.

## Neutrophil function

Trauma initiates a 'stress response' through the endocrine, metabolic and inflammatory systems. The primary endocrine response is to produce catecholamines and corticosteroids, raising the immune response and mobilisation of neutrophils.[68] Neutrophil function will be analysed through the validated PhagoBURST and PhagoTest kits (Glycotope Biotechnology GmbH, Germany) to assess superoxide generation and phagocytosis, respectively. This gives a pilot opportunity to test whether DHEA supplementation improves the immune response and, in turn, has the potential to protect against infection. However, benefits might only be detectable after a period of DHEA supplementation that is substantially longer than 3 days.

## Proinflammatory and anti-inflammatory cytokines

Prolonged CARS may leave the recovering patient susceptible to increased risk of late infection.[69] The cytokine storm of interleukin (IL) 6 and IL-10 have demonstrated a strong association with the severity of injury and mortality[70] but less so sepsis.[71] These post-injury cytokines are also known to affect the peripheral target tissues that are involved in steroid metabolism.[72] The post-injury proinflammatory and anti-inflammatory cytokines assessed have been selected based on previous work from our group.[73]

## Tolerance: gastric residual volume (GRVs)

Swallowing difficulties, facial injuries or a non-functioning gut may prohibit sublingual or oral administration and compliance to the study protocol. Therefore, an adaptable study design is needed to generate pilot data. By monitoring GRVs, a surrogate marker of gastrointestinal motility,[74] we will regard a GRV persistently exceeding 250 mL as intolerable.

## Treatment compliance and evaluability

To meet study compliance and be considered evaluable, the following must be satisfied:
► Patients must be sufficiently dosed on at least 1 day of DHEA administration.
► If a patient fails to consume the intended IMP or vomits within 1 hour of consuming the IMP, this dosing will be classed as *insufficient.*

## Data Monitoring Committee (DMC)

Data analyses will be supplied in confidence to an independent DMC, which will be asked to advise on whether the accumulated data from the trial, together with the results from other relevant research, justify the continuing recruitment of further patients. The DMC will operate under a trial-specific charter based on the template created by the Damocles Group.

Results will be provided to the DMC and discussed via teleconference at a minimum. In consultation with the trial statistician, the DMC will meet when any cohort undergoes a dose escalation decision. The DMC will advise on dose escalation based on the rules as described

previously. Additional meetings may be called if recruitment is much faster than anticipated, and the DMC may, at their discretion, request to meet more frequently or continue to meet following completion of recruitment. An emergency meeting may also be convened if a safety issue is identified. The DMC will report directly to the Trial Management Group. The DMC may consider recommending the discontinuation of the trial if the recruitment rate or data quality is unacceptable or if any issues are identified which may compromise patient safety. The trial would also stop early if the interim analyses showed differences between treatments that were deemed to be convincing to the clinical community. Data monitoring members have undertaken their initial review of the first 16 patients on the 8 November 2019. The outcome of this DMC required all-female cohorts to continue 50 mg of DHEA (both the sublingual and oral), with both male cohorts increasing to 100 mg.

## Statistical analysis

### Sample size

The maximum sample size will be 270 (6 groups of 15 participants being administered 3 different dose levels). However, we predict the realised sample size to be smaller as there are likely to be early opportunities to escalate dose level within a group. Following consultation with the trial statistician, 15 participants were chosen to provide a modest amount of information on the primary outcome at each dose in each group while allowing recruitment to be completed in a reasonable amount of time. Statistical power calculation has not been performed as ADaPT is a dose escalation trial, and we are not applying a null hypothesis significance testing approach.

### Primary outcome

Serum DHEA concentrations will be summarised as means and SDs (or medians and IQRs, if non-normal) at each time point and dose in each cohort, where a cohort is the three-way combination of patient cohort, dose and administration route. The observed rate of DHEA normalisation will be reported for each cohort with CIs calculated using Wilson's method. The cohorts sample sizes are small, so these cohort-specific analyses are likely to be relatively imprecise. However, the total sample size is large for the trial phase, and there is much information in the cohort structure that will likely be pertinent to understanding the outcomes. Supplementary analyses to support dose decisions will be provided using hierarchical regression models that analyse outcomes from all cohorts and doses together while reflecting cohort memberships. These models are explained further.

### Modelling serum DHEA concentrations and DHEA normalisation probability

We propose hierarchical Bayesian models to analyse serum DHEA outcomes in all cohorts simultaneously. An intercept will be included to estimate the mean population-level response common to all cohorts plus further terms

to reflect effects for dose, patient type and administration method. Further population-level terms (also called fixed effects) will be considered, including baseline DHEA level and age. Interactions will be considered as appropriate. Group-level terms (also called random effects) to reflect cohort and patient heterogeneity will be considered.

Modestly informative or regularising priors will be used that anticipate little or no effect (i.e, have a mean close to or equal to zero) but restrict the range of parameter values to those that are feasible (i.e, do not place undue or unrealistic probability mass in wide distribution tails). Such priors can be considered informative of scale but not location in that they do not anticipate effects but rule out infeasible values. They are effective at dissuading models from overfitting and aiding convergence in the posterior sampler when there are many parameters. The leave-one-out cross validation (LOO) or widley applicable information criteria (WAIC) will be used to find parsimonious but informative models.

It is anticipated that: the probability of DHEA normalisation will be modelled using a generalised linear model with logit link function, and postbaseline DHEA will be modelled using a generalised linear model with identity or log (for positive data) link functions.

## INTERIM ANALYSIS

There will be an interim analysis presented when any cohort undergoes a dose-escalation decision, as previously described. The particular objective of this analysis will be to assess if the rate of DHEA normalisation is too low and whether there is evidence that motivates investigating a higher dose in that cohort. The primary and supporting analyses of the primary outcome will be presented, as described previously.

## ETHICS AND DISSEMINATION

This protocol has been approved by a Research Ethics Committee (Reference 18/WM/0102) on 8 June 2018. Results will be disseminated via peer-reviewed publications and presented at national and international conferences. This will be coordinated with members of the research team and the patient and public involvement group. The study investigator is responsible for communicating important protocol modifications to relevant parties.

## TRIAL PROGRESSION

The study recruited its first patient on 2 April 2019 and held its first data monitoring committee on 8 November 2019. Currently, all female groups remain on 50 mg DHEA, whilst both male cohorts have increased to 100 mg. The dose escalation also coincided with the first sponsor audit of ADaPT. Site audits will occur at times of escalation and interim analysis until the study is completed.

**Author affiliations**
[1] NIHR Surgical Reconstruction and Microbiology Research Centre, Queen Elizabeth Hospital Birmingham, Birmingham, UK

[2] School of Sport, Exercise and Rehabilitation Sciences, University of Birmingham, Birmingham, UK
[3] D3B, CRUK Clinical Trials Unit, University of Birmingham College of Medical and Dental Sciences, Birmingham, UK
[4] Institute of Metabolism and Systems Research, University of Birmingham College of Medical and Dental Sciences, Birmingham, UK
[5] Institute of Inflammation and Ageing, University of Birmingham, Birmingham, UK
[6] NIHR Birmingham Liver Biomedical Research Unit Clinical Trials Group (D3B team), CRUK Clinical Trials Unit, University of Birmingham, Birmingham, UK
[7] MRC-Versus Arthritis Centre for Musculoskeletal Ageing Research, Birmingham, UK
[8] National Institute of Health Research, Birmingham Biomedical Research Centre, University Hospitals Birmingham NHS Foundation Trust and University of Birmingham, Birmingham, UK
[9] Royal Centre for Defence Medicine, Queen Elizabeth Hospital Birmingham, Birmingham, UK

**Acknowledgements** The authors would like to thank the patients, relatives, clinical staff, research nurses and support staff in the National Institute for Health Research Surgical Reconstruction and Microbiology Research Centre (NIHR-SRMRC) at the Queen Elizabeth Hospital for their continued support for the study. NIHR-SRMRC clinical research team members consulted at the beginning of the study are: Gurneet Sur, Victoria Homer, Lauren Cooper; Morgan Foster, Chris McGhee, Colin Bergin, Amy Bamford, Karen Ellis, Emma Fellows, Stephanie Goundry, Elaine Spruce, Katie Moss, Tracy Mason, Christina Bratten, Liesl Despy, Samantha Harkett, Yin May, Natalie Dooley and Hazel Smith.

**Contributors** CB, KB, VH and CP have prepared the manuscript. CB, CP, MAF, WA, JL, CAG, AT, LG, JH, KB, AA, DB, RC, GS and KY were involved in the methodological design and drafting of the trial protocol. JL, CB, CE and KM validated laboratory equipment and sample analysis for PB and PT testing. AD undertook all aspects are pharmacy procedure and IMP negotiations. KMY, ET, MAF, RC and CB enrolled participants and assisted with data collection and study-specific procedures. LG, AT, FS and WA undertook the LC-MS validation and processing of patient sex steroids samples. KB and VH are the trial statisticians who designed and wrote the analysis plan and code for randomisation of patients and times. CP, AA, DB and GS are the trial coordinators. All authors reviewed and edited the manuscript.

**Funding** For funding this research, the authors acknowledge the NIHR-SRMRC, AOUK&I Foundation and the Queen Elizabeth Hospital Charity Birmingham. Trial sponsor: Research & Development (Governance), University Hospitals Birmingham NHS foundation Trust, Heritage Building, Queen Elizabeth Hospital, Mindlesohn Way, Edgbaston, Birmingham, B15 2GW.

**Competing interests** None declared.

**Patient and public involvement** Patients and/or the public were involved in the design, or conduct, or reporting, or dissemination plans of this research. Refer to the Methods section for further details.

**Patient consent for publication** Not required.

**Provenance and peer review** Not commissioned; externally peer reviewed.

**ORCID iDs**
Conor Bentley http://orcid.org/0000-0003-3021-4838
Kamal Makram Yakoub http://orcid.org/0000-0001-7697-2453
Emma Toman http://orcid.org/0000-0003-2142-1923
Jon Hazeldine http://orcid.org/0000-0002-4280-4889
Janet M Lord http://orcid.org/0000-0003-1030-6786

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
