## [Reviewer comments · BMJ Open]

ARTICLE DETAILS

TITLE (PROVISIONAL)	A prospective, phase II, single-centre, cross-sectional, randomised study investigating Dehydroepiandrosterone supplementation and its Profile in Trauma: ADaPT
AUTHORS	Bentley, Conor; Potter, Claire; Yakoub, Kamal; Brock, Kristian; Homer, Victoria; Toman, Emma; Taylor, Angela; Shaheen, Fozia; Gilligan, Lorna; Athwal, Amrita; Barton, Darren; Carrera, Ronald; Young, Katie; Desai, Amisha; McGee, Kirsty; Ermogenous, Christos; Sur, Gurneet; Greig, Carolyn; Hazeldine, Jon; Arlt, Wiebke; Lord, Janet; Foster, Mark

VERSION 1 – REVIEW

REVIEWER	Aoki, Makoto Gunma University
REVIEW RETURNED	05-Sep-2020

GENERAL COMMENTS	Thank you for giving me the opportunity to review the manuscript. The manuscript was protocol paper of RCT (A prospective, phase II, single-centre, cross-sectional, randomised study investigating Dehydroepiandrosterone supplementation and its Profile in Trauma: ADaPT) As reviewer, I was interested in the use of dehydroepiandrosterone supplementation for trauma patients. Regarding protocol, I had a few questions. #1. The exclusion criteria of the patients aged 16-50. The authors excluded isolated TBI patients and did not exclude the non-isolated TBI patients. Non isolated TBI patients were better excluded? I thought TBI patients could affect the hormone levels. #2. Regarding statistical analysis, the authors did not perform power calculation. As authors pointed out this study divided into six groups and each group size (n=15) was small. Even if the null hypothesis was not made, the authors set primary outcome and power calculation was needed? And final statistical analysis was performed putting together the patients aged 16-50 and >50 years? I thought the difference of population age could strongly affect the hormone levels and statistical analysis had better be divided in each population.
--

REVIEWER	Skene, Simon University of Surrey, Surrey Clinical Trials Unit
REVIEW RETURNED	05-Dec-2020

GENERAL COMMENTS	This is a clearly written protocol, and objectives and outcomes well described. I have some minor comments on the statistical matters which I think could be clarified to improve the paper. 1. Whilst power is not addressed directly, the assumption of 15 participants evaluated in each cohort with an expectation of 13/15 'successes' (approx 85%) does suggest the acceptable precision the trial team are prepared accept in calling the primary outcome as being achieved. I think this would be useful; to note, with a calculation of the precision. 2. Escalation between doses is allowed. Again I think it would be useful to outline criteria for the DMEC or similar to allow this, and for de-escalation. Is a 'playbook' to be prepared in advance, or are pre-specified analyses points within each cohort defined? Presumably separate discussions will beheld for the male/female cohorts to allow for differences?
--

VERSION 1 – AUTHOR RESPONSE

Reviewer 1 comments:

The exclusion criteria of the patients aged 16-50. The authors excluded isolated TBI patients and did not exclude the non-isolated TBI patients. Non isolated TBI patients were better excluded? I thought TBI patients could affect the hormone levels.

Author's comments:

Thank you for your comment. This is correct. Patients who have received an isolated TBI have not been included in this study. However, those patients who have sustained a polytraumatic injury that consists of a TBI will still be enrolled. However, the TBI in the polytrauma patient cannot be considered the primary injury, involve the HPA axis and must be minor compared to the other injuries sustained. A consultant must evaluate all patients before study entry, and any patients who are considered borderline will not be entered.

Reviewer 1 comments:

4. Regarding statistical analysis, the authors did not perform power calculation. As authors pointed out this study divided into six groups and each group size (n=15) was small. Even if the null hypothesis was not made, the authors set primary outcome and power calculation was needed? And final statistical analysis was performed putting together the patients aged 16-50 and >50 years? I thought the difference of population age could strongly affect the hormone levels and statistical analysis had better be divided in each population.

Author's comments:

Thank you for your comments. ADaPT is an early phase, dose-escalation trial where no hypotheses regarding the primary outcome have been made a priori and indeed where no formal hypothesis testing will be carried out in that we will not be formally comparing outcomes either within patient cohort or between them. The choice of n=15 per group was selected to provide equipoise between the information required to draw accurate conclusions and what was feasible to recruit during the recruitment period. The decision to dose escalate and the criteria for concluding success from the trial are principally rule based (a setting where it is unconventional to perform power calculations) with the described hierarchical modelling used to supplement and support these decisions. For dose escalation, this rule-based analysis will provide an early opportunity to escalate when a probabilistic analysis will likely be under-informed due to the small sample size. The results of this trial are likely to

inform a larger phase III confirmatory trial of the final chosen dose against standard of care to determine superiority of DHEA, where the necessary power calculations will be performed. During all analyses, patients will be analysed as per their cohort, where a cohort is the three-way combination of patient characteristics (e.g. male trauma, female trauma, or female hip fracture), dose and dose-administration route, and thus has a maximum sample size of 15. In the proposed hierarchical modelling all patients data will contribute to the same model, however there will be covariates to indicate the cohort they originated from and conclusions drawn from the modelling will again be on a cohort level. Modifications to the paper have been made to clarify both of the above.

Reviewer 2 comments:

5. Whilst power is not addressed directly, the assumption of 15 participants evaluated in each cohort with an expectation of 13/15 'successes' (approximately 85%) does suggest the acceptable precision the trial team are prepared accept in calling the primary outcome as being achieved. I think this would be useful; to note, with a calculation of the precision.

Author's comments:

Thank you for your comments. As outlined above in response to reviewer 1's comments, no sample size calculations have been performed as this is an early phase, rule-based dose escalation trial (with supporting modelling) with no formal hypothesis tests planned, therefore any indication of precision is hard to gauge. The sample size of 15 per group was based on feasibility to recruit while provide sufficient necessary information on the outcomes and the requirement of 13/15 successes chosen as this is what was thought to be clinically relevant to warrant further investigation in a larger confirmatory phase III trial to ascertain superiority of DHEA compared to standard of care. While the individual estimates will be imprecise, we hope the explained Bayesian hierarchical modelling will be able to offer slightly more precise estimates due to the pooling of data and nature of the proposed analysis. We have added comments regarding the choice of 13 and 15 to the paper.

Reviewer 2 comments:

6. Escalation between doses is allowed. Again I think it would be useful to outline criteria for the DMEC or similar to allow this, and for de-escalation. Is a 'playbook' to be prepared in advance, or are pre-specified analyses points within each cohort defined? Presumably separate discussions will be held for the male/female cohorts to allow for differences?

Author's comments:

Thank you for your comments, we have amended the text regarding dose escalation rules and how the DMC are to make these decisions accordingly. There are no pre-defined de-escalation rules as it is anticipated that there will be no toxicity concerns (as DHEA is licensed as a food supplement in the US) and therefore the risk of over-dosing is minimal. Any decision to de-escalate would be at the discretion of the DMC made on the judgement of the safety information presented. The DMEC meetings are not necessarily separate for each cohort (as all groups are recruited to in parallel and the required mass spectrometry to ascertain the primary outcome is done in batches), however the outcomes are presented and decisions made separately for each cohort. Wording has been added to make it clear outcomes and thus escalation decisions are considered separately for each cohort.

VERSION 2 – REVIEW

REVIEWER	Skene, Simon University of Surrey, Surrey Clinical Trials Unit
REVIEW RETURNED	15-Jun-2021
GENERAL COMMENTS	The authors have addressed my minor concerns. I would still contend that the decision to pick 15 per cohort with a (clinically relevant) success threshold of 13 implies an acceptable precision for decision making. Whilst I accept that the authors did

	not base this on an a priori power calculation, the ultimate success of the trial and decision to proceed to a larger RCT will be judged on this basis. It may be beneficial to a future publication of results to clarify this, but I am content to leave that as a matter for the authors.
--	---